# Diagnosis of human brucellosis: Systematic review and meta-analysis

**Mariana Lourenço Freire**[1]☯*, **Tália Santana Machado de Assis**[1,2]☯, **Sarah Nascimento Silva**[1], **Gláucia Cota**[1]

**1** Grupo de Pesquisa Clínica e Políticas Públicas em Doenças Infecciosas e Parasitárias, Instituto René Rachou, Fundação Oswaldo Cruz, Belo Horizonte, Brasil, **2** Centro Federal de Educação Tecnológica de Minas Gerais, Contagem, Minas Gerais, Brasil

☯ These authors contributed equally to this work.
* marianalfreire@hotmail.com

**Data Availability Statement:** All relevant data are within the manuscript and its Supporting Information files.

**Funding:** This work was supported by the Conselho Nacional de Desenvolvimento Científico e

## Abstract

### Background

Brucellosis, a widely spread zoonotic disease, poses significant diagnostic challenges due to its non-specific symptoms and underreporting. Timely and accurate diagnosis is crucial for effective patient management and public health control. However, a comprehensive comparative review of available diagnostic tests is lacking.

### Methodology/Principal findings

This systematic review addressed the following question: 'What is the accuracy of the available tests to confirm human brucellosis?' Two independent reviewers examined articles published up to January 2023. The review included original studies reporting symptomatic patients with brucellosis suspicion, through any index test, with sensitivity and/or specificity as outcomes. As exclusion criteria were considered: sample size smaller than 10 patients, studies focusing on complicated brucellosis, and those lacking essential information about index or comparator tests. Sensitivity and specificity were assessed, with consideration for the index test, and 'culture' and 'culture and standard tube agglutination test (SAT)' were used as reference standards. Bias assessment and certainty of evidence were carried out using the QUADAS-2 and GRADE tools, respectively. A total of 38 studies reporting diagnostic test performance for human brucellosis were included. However, the evidence available is limited, and significant variability was observed among studies. Regarding the reference test, culture and/or SAT are deemed more appropriate than culture alone. Rose Bengal, IgG/IgM ELISA, and PCR exhibited equally high performances, indicating superior overall diagnostic accuracy, with very low certainty of the evidence.

### Conclusions/Significance

This systematic review underscores the potential of the Rose Bengal test, IgG/IgM ELISA, and PCR as promising diagnostic tools for brucellosis. However, the successful implementation and recommendations for their use should consider the local context and available resources. The findings highlight the pressing need for standardization, improved reporting,

Tecnológico (301384/2019 to GC; 151891/2022-2 to MLF) and Ministry of Health - Brazil (TED 20/2022 to Fiocruz). The funders had no role in study design, data collection and analysis, decision to publish, or preparation of the manuscript.

**Competing interests:** The authors have declared that no competing interests exist.

and ongoing advancements in test development to enhance the accuracy and accessibility of brucellosis diagnosis.

## Author summary

Brucellosis represents a prevalent zoonotic condition that significantly impacts regions constrained by limited resources. Diagnosis, usually based on symptoms and incomplete data, leads to underreporting and delayed treatment. Our comprehensive systematic literature review focused on evaluating the practical effectiveness of current diagnostic approaches for brucellosis, to guide decision-makers. Our analysis involved 38 studies primarily conducted in Asian and African regions, revealing considerable outcome variability. When considering the reference test, culture and/or SAT are deemed more suitable than culture alone. While Rose Bengal, IgG/IgM ELISA, and PCR tests exhibited equally strong performances, the evidence remained notably limited. It is vital to recognize that apart from performance, factors such as accessibility, cost, and ease of use must also be factored into informed decision-making. Our findings emphasize the critical need to expand the scope of validation studies on diagnostic tests and the development of new, more robust, and easily accessible alternatives for addressing brucellosis. This pursuit is essential to meet the urgent demand for enhanced diagnostic capabilities in this field, providing improved methods to combat this challenging disease.

## Introduction

Brucellosis is a reemerging and neglected zoonotic disease caused by facultative intracellular Gram-negative coccobacilli belonging to the genus *Brucella*. It is considered the most prevalent bacterial disease worldwide, with endemicity primarily observed in regions spanning the Middle East, Asia, Africa, South and Central America, the Mediterranean Basin, and the Caribbean [1,2]. Accurate measures of disease burden are hampered by a substantial number of asymptomatic cases, difficulties in reaching a definitive diagnosis, and underreporting [3].

Clinically, brucellosis can manifest as an acute or insidious disease, presenting a variable pattern of fever, malaise, and night sweats. Additional symptoms include weight loss, arthralgia, headache, lower back pain, fatigue, anorexia, myalgia, cough, and emotional changes with a depressive tendency [4–6]. If left untreated, the disease can become prolonged, lasting for months to years. Due to the non-specific and broad clinical spectrum of brucellosis, it is essential to differentiate it from other infectious and non-infectious diseases, including typhoid fever, malaria, bacterial arthritis and endocarditis, tuberculosis, pneumonia, rheumatologic diseases, neoplasms, fungal infections, and psychiatric disorders [7–9]. Confirmatory diagnosis of brucellosis traditionally requires *Brucella* sp. isolation in culture of clinical specimens such as blood or other body fluids [10,11]. Although widely accepted, culture isolation requires substantial time and is not generally very sensitive, especially during the later stages of the disease [12,13]. In addition, culture-based diagnosis is labor-intensive and comes with a significant risk of contamination to the professionals who handle the samples, requiring biosafety cabinets for manipulation [14]. Immunological diagnostic methods therefore became more widely used, especially the Rose Bengal test, the standard tube agglutination test (SAT), enzyme-linked immunosorbent assays (ELISA), the Coombs test, and immunochromatographic tests [10,11]. Immunological tests are inexpensive and user friendly but have as the

main limitations the lack of common interpretative criteria and the suboptimal specificity due to interspecies cross-reactivity. The performance of many minimally different tests using the same platform are available in medical literature. However, the diversity of scenarios, cut-offs and populations hinder a comparative view of the results to support the definition of a diagnostic algorithm [10,15].

Recently, molecular methods, such as qualitative and quantitative polymerase chain reaction (PCR) targeting various genes, have also been utilized for diagnosing this disease [16,17]. As in other diseases, molecular tests are expected to have high sensitivity, however, they may not necessarily indicate an active infection but rather a low bacterial inoculum, DNA from dead bacteria, or a past infection. In the same way as for serology, the wide variation in reagents, targets, and methods, specially between commercial and home-made molecular tests, make the systematic collection of data, followed by a critical analysis, the first step in staging the available level of evidence [15,18].

Although various strategies are employed and many studies have been published on diagnosing human brucellosis, the systematic gathering of data and a comparative and critical analysis of factors determining the performance of distinct tests have not been undertaken. The present study aimed to systematically review the literature to summarize the evidence on the diagnostic accuracy (sensitivity and specificity) of the diagnostic tests available for diagnosing human brucellosis.

## Methods

### Protocol and registration

This systematic review was conducted in accordance with the methodological principles given in the Cochrane Handbook [19] and it adhered to the Preferred Reporting Items for Systematic Reviews and Meta-analysis (PRISMA) guidelines [20]. The study protocol was registered in PROSPERO (CRD42023411933).

### Eligibility criteria

The following research question was formulated to guide the systematic review: 'What is the accuracy of the available tests to confirm human brucellosis?' The article selection process adhered to the PICOS framework (population, intervention, comparator, outcome, study design), and the inclusion criteria were as follows: (P) symptomatic patients presenting with suspected brucellosis, (I) any diagnostic test (index test), (C) culture or other specified diagnostic tests for comparison, (O) sensitivity and/or specificity as the primary outcomes, and (S) original studies focused on diagnostic accuracy.

The exclusion criteria included studies with a sample size of fewer than 10 patients (cases with a confirmed diagnosis of brucellosis), those that presented patients with localized/complicated forms of human brucellosis, those lacking essential information regarding the index test or reference standard, or studies published in languages other than English, Spanish, or Portuguese.

### Search strategy

Systematic literature searches were conducted in four databases: MEDLINE (PubMed), Embase, Cochrane Central Registry of Controlled Trials (CENTRAL), and the Virtual Health Library (VHL). For each database, keywords related to 'human brucellosis,' 'diagnosis,' and 'sensitivity or specificity' were combined with Boolean operators (AND, OR). The S1 File provides a detailed description of the search strategy used for each database. Articles published up

to January 6, 2023, were included, with no restrictions on publication date. Additional searches of the reference lists of the articles included were also conducted.

## Selection process

Records obtained from each database were imported into Mendeley Reference Management to identify and eliminate duplicate files [21]. Subsequently, these records were imported into Rayyan for title and abstract screening [22]. The screening process was carried out independently by two reviewers (MLF, TSMA) following predefined inclusion and exclusion criteria. Disagreements were resolved through consensus, and in cases where consensus could not be reached, two additional reviewers (SNS, GFC) were invited for resolution. The full texts of selected studies were read to confirm their eligibility, extract pertinent data, and verify that exclusion criteria did not apply.

## Data extraction

A pivotal step in this study was the definition of the reference standards to be used for analysis. Despite being historically recognized as the traditional reference test due to its high specificity, culture isolation is considered an imperfect reference test due to its estimated low sensitivity. This fact explains why most of the reviewed studies combined culture isolation and/or SAT as the criterion for defining a 'true brucellosis case.' In this systematic review, both 'culture' and 'culture or SAT' were considered as reference tests. The greater number of studies gathered and the lower heterogeneity between studies using 'culture and/or SAT" as a reference test confirmed that this was the most suitable approach for the proposed analysis.

The main study characteristics, information related to the population, intervention, comparator, and outcome were extracted from all articles by two reviewers. Data of interest included the country in which the study was conducted, the diagnostic methods used, the number of participants tested, the reference test, study design, onset time of symptoms, age, gender, *Brucella* species involved, and the characteristics of the tests, including manufacturers, antigens, and titers used for defining test positivity. To compute sensitivity and specificity values, whenever possible, raw data from primary studies were extracted to populate the four cells of a 2×2 diagnostic table: true positives, false positives, true negatives, and false negatives. A second researcher independently verified the extraction of primary data from each study.

## Data synthesis and statistical analysis

For each diagnostic test identified, the pooled sensitivity and specificity were estimated considering both 'Culture' and 'Culture and/or SAT' as the reference standard. For this, CMA version 3.0 was used. For all analysis, the random-effects model was used, an approach that account for heterogeneity among studies resulting in wider confidence intervals and less precise central performance estimate measurements.

When the same index test was evaluated more than once in the same study, either by different manufacturers or using the same test with different cut-off points, we only considered the result that gave the best performance. Different results were presented for the same patients using plasma or total blood; those which used the more frequently used sample type in the other studies were chosen for inclusion in the pooled analysis.

## Risk of bias

The risk of bias in primary studies was independently assessed by two reviewers (MLF and TSMA) using the QUADAS-2 tool [23]. This tool comprises four key domains for assessing

the risk of bias (patient selection, index test, reference standard, and flow and timing). The first three of those domains were also used to assess the applicability of the systematic review. Carefully selected signaling questions were employed to guide evaluations across all domains.

### Certainty of evidence

The quality of evidence for the optimal diagnostic accuracy tests was evaluated using the GRADE tool [24]. This tool provides a guide for assessment targets on various factors that may potentially reduce the quality of evidence, including the risk of bias, indirect evidence, inconsistency, imprecision, and publication bias. With this tool, the quality of evidence can be categorized into four levels: high, moderate, low, and very low.

## Results

### Literature search

The initial search identified 4,379 articles in the databases. Following screening and selection based on eligibility criteria, 90 studies were selected for potential for inclusion. To perform an indirect comparison among tests, studies were grouped according to the reference test adopted, which could be 'culture' or 'culture or SAT.' Studies that used other reference tests were excluded, resulting in the inclusion of 38 studies. The PRISMA flow diagram summarizes the study selection process and reasons for exclusion (Fig 1).

### Descriptive analysis of included studies

The main characteristics of the studies included are summarized in Table 1. Most studies were conducted in Asian countries (24/30), while studies in Europe and the Americas were scarce (five and two studies, respectively). In eight studies, the origin of the patients was not reported. Among the included studies, ELISA was the most frequently evaluated index test (55%; 21/38), followed by SAT (34.2%; 13/38), PCR (31.6%; 12/38), and Rose Bengal (28.9%; 11/38). The immunochromatographic test was assessed in only four studies, and the Coombs test in five. Overall, the included studies were predominantly retrospective (60.5%; 23/38), and the main characteristics of the population studied, such as age (52.6%; 20/38) and gender (52.6%; 20/38), as well as duration of the disease (63.2%; 24/38), were frequently not available.

### Summary of results

Table 2 presents pooled sensitivity and specificity measures for the index tests, considering both 'culture' and 'culture or SAT' as reference standards. Details about each included article for every index test are provided in the S2 File. This comparative approach supported selecting the combined criterion 'culture or SAT' as a less flawed reference standard for brucellosis. This choice was based on the poorer specificity exhibited by all index tests (overestimated by an insensitive comparator) and greater heterogeneity among studies for most analyses when culture was utilized as the reference test (as depicted in the S3 File).

### Serological tests

Three studies assessed a non-titratable agglutination test using the Rose Bengal antigen [56,58,61]. The summarized measures of sensitivity and specificity were 96.6% [95% CI: 92.6–98.5] and 97.9% [95% CI: 93.1–99.4], respectively ($I^2 = 0$) (Fig 2).

Seven studies assessed the presence of the antibody subclasses IgM and IgG, detected by *ELISA*. Tests based on the presence of IgG or IgM subclasses exhibited the highest

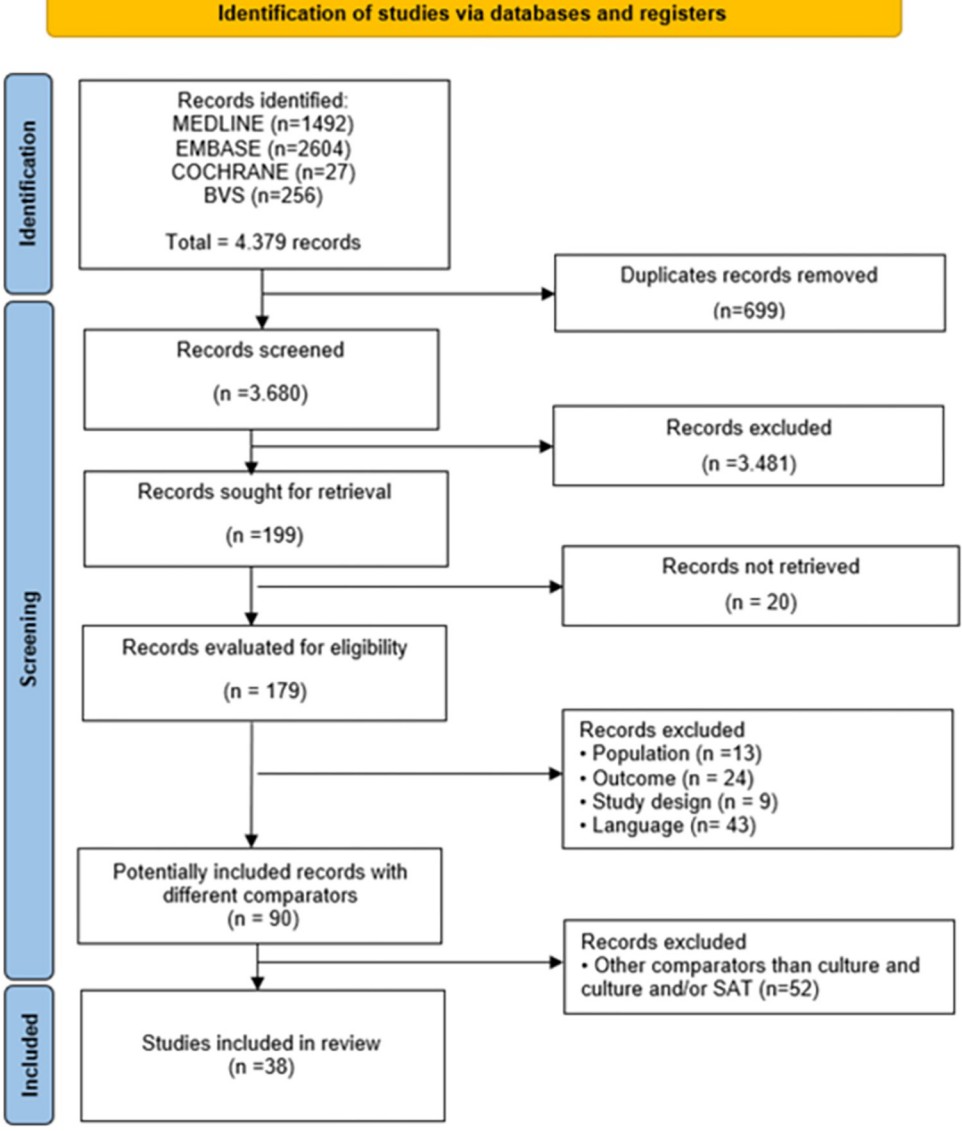

**Fig 1. Flow diagram illustrating the study selection process according to PRISMA.**

performance: 96.8% [95% CI: 60.8–99.8; $I^2$ = 88.90] and 98.6% [95% CI: 96.1–99.5; $I^2$ = 0.0], for sensitivity and specificity, respectively (Fig 3).

Two studies [56,61] reported the performance of immunochromatographic tests for brucellosis, considering culture and/or SAT as the reference standard. Only sensitivity for IgM could be estimated (70.6% [95% CI: 62.9–77.3; $I^2$ = 0.00]) (Fig 4).

Two studies [34,43] were found comparing the Coombs test with culture and/or SAT. Pooled sensitivity and specificity were 89.4% [95% CI: 30.0–99.4; $I^2$ = 73.6] and 98.8% [95% CI: 92.0–99.8; $I^2$ = 0.00], respectively (Fig 5).

## Molecular test

The accuracy of qualitative PCR was evaluated in four studies, with a pooled sensitivity of 79.6% [95%CI: 47.6–94.4; $I^2$ = 90.4] and specificity of 96.0% [95%CI: 85.8–99.0; $I^2$ = 47.8]

**Table 1. The main characteristics of the studies included on the diagnosis of human brucellosis.**

| Reference | Origin of patients | Reference Test | Index test | Sample size (case/non case) | Study design | Onset symptom in months (mean/median) | Age (mean/median) | Male/Female | *Brucella* species identified (n) |
|---|---|---|---|---|---|---|---|---|---|
| **Nicoletti et al., 1971 [25]** | Iran | Culture | 1) SAT; 2) Rose Bengal | 16/196 | Prospective | NR | NR | NR | *B. melitensis* (1) |
| **Kiel et al., 1987 [26]** | Saudi Arabia | Culture | SAT | 60 | Retrospective | NR | 32 | 35/25 | *B. melitensis; B. abortus* |
| **Saz et al., 1987 [27]** | Spain | Culture | 1) SAT; 2) Rose Bengal; 3) ELISA; 4) Coombs | 208/107 | Retrospective | NR | NR | NR | *B. melitensis* |
| **Araj et al., 1988 [28]** | Kuwait | Culture | 1) Rose Bengal; 2) SAT; 3) ELISA | 83/72 | Retrospective | < 2 | NR | NR | *B. melitensis* (83) |
| **Araj et al., 1990 [29]** | Kuwait | Culture | ELISA | 21/15 | Retrospective | < 2 | 31 (17–59) | NR | *B. melitensis* |
| **Queipo-Ortuño et al., 1997 [30]** | Spain | Culture | Conventional PCR | 35/60 | Prospective | 0.85 (0–4) | NR | 37/10 | *B. melitensis* (35) |
| **Osoba et al., 2001 [31]** | NR | Culture | ELISA | 30/44 | Retrospective | < 2 meses | NR | NR | *B. melitensis* (30) |
| **Memish et al., 2002 [32]** | Saudi Arabia | Culture | 1) SAT; 2) ELISA | 68/70 | Prospective | NR | NR | NR | |
| **Mert et al., 2003 [33]** | Turkey | Culture | 1) SAT; 2) Rose Bengal | 30/280 | Retrospective | NR | NR | NR | NR |
| **Vrioni et al., 2004 [34]** | Greece | Culture | PCR-ELISA | 179 | Retrospective | 1.08 (0.23–2) | 46.1 (18–91) | 163/80 | *B. melitensis* (179) |
| **Al-Nakkas et al., 2005 [35]** | Kuwait | Culture | Conventional PCR | 89/244 | Prospective | NR | NR | NR | *B. melitensis* (85) / *B. abortus* (4) |
| **Debeaumont et al., 2005 [36]** | NR | Culture | Real-time PCR | 17/60 | Retrospective | NR | 45 (11–69) | 10/7 | NR |
| **Ertek et ak., 2006 [37]** | NR | Culture | 1) SAT 2) ELISA | 32/20 | Retrospective | NR | NR | NR | NR |
| **Fadeel et al., 2006 [38]** | Egypt | Culture | ELISA | 202/103 | Retrospective | 0.33 (0.1–3) | 28 (3–60) | 144/58 | NR |
| **Abdoel et al., 2007 [39]** | NR | Culture | 1) SAT; 2) Rapid test; 3) Coombs | 45 | Retrospective | NR | NR | NR | *B. melitensis* (45) |
| **Mizanbayeva et al., 2009 [40]** | Kazakhstan | Culture | 1) SAT; 2) Rose Bengal; 3) Rapid test | 63 | Retrospective | < 6 (n = 50) 6–12 (n = 2) > 12 (n = 11) | 33 (21–83) | 46/17 | NR |
| **Mantur et al., 2010 [41]** | India | Culture | ELISA | 31/72 | Prospective | NR | 27.46 ± 18.74 | 03/01 | *B. melitensis* |
| **Al-Ajlan et al., 2011 [42]** | Saudi Arabia | Culture | 1) conventional PCR; 2) real-time PCR | 89/40 | Retrospective | NR | NR | NR | *Brucella spp.* |
| **Díaz et al., 2011 [43]** | Spain | Culture | 1) SAT; 2) Rose Bengal | 208/20 | Retrospective | NR | NR | NR | *B. melitensis* |
| **Peeridogaheh et al., 2013 [44]** | NR | Culture | 1) ELISA; 2) Coombs | 11/32 | Retrospective | NR | NR | NR | NR |
| **Ayala et al., 2014 [45]** | NR | Culture | ELISA | 49/77 | Retrospective | NR | NR | NR | *Brucella spp.* |
| **Purwar et al., 2016 [46]** | India | Culture | 1) SAT; 2) Rose Bengal | 20/360 | Cross-sectional | NR | NR | NR | NR |
| **Akhvlediani et al., 2017 [47]** | Georgia | Culture | 1) SAT; 2) ELISA | 33/48 | Prospective | NR | 39.9 (DP 15,1) | 60/21 | *B. melitensis* (32) / *B. abortus* (1) |
| **Dal et al., 2018 [48]** | Turkey | Culture | Real-time PCR | 36/117 | Retrospective | < 2 (n = 210) / > 3 (n = 5) | 30 (2–80) | 92/123 | *B. melitensis* |

*(Continued)*

**Table 1.** (Continued)

| Reference | Origin of patients | Reference Test | Index test | Sample size (case/non case) | Study design | Onset symptom in months (mean/median) | Age (mean/median) | Male/Female | *Brucella* species identified (n) |
|---|---|---|---|---|---|---|---|---|---|
| **Patra et al., 2019** [49] | India | Culture | Real-time PCR | 53/54 | Prospective | 1.23 (IQ: 0.77–2) | NR | NR | *B. melitensis* (53) |
| **Xu et al., 2020** [50] | China | Culture | 1) SAT; 2) ELISA | 51/248 | Prospective | NR | 48.39 ± 19.96 | 28/23 | NR |
| **Zhao et al., 2020** [51] | China | Culture | Real-time PCR | 46/62 | Retrospective | NR | NR | NR | *B. melitensis* |
| **Almashhadany et al., 2022** [52] | Iraq | Culture | Rose Bengal | 31/297 | Cross-sectional | NR | 44 (18–82) | 172/153 | *B. melitensis* (18) / *B. abortus* (13) |
| **Nimri, 2003** [53] | Jordan | Culture; Culture and/or SAT | Conventional PCR | 20/25; 140/25 | Prospective | 1.16 (0.33–2) | 46 (6–86) | 58/107 | NR |
| **Al-Shamahy et al., 1998** [54] | NR | Culture and/or SAT | ELISA | 146/20 | Retrospective | NR | NR | NR | *B. melitensis* |
| **Al-Attas et al., 2000** [55] | Saudi Arabia | Culture and/or SAT | Conventional PCR | 14/32 | Prospective | 4 (1–13) | 6 a 65 | 8/7 | NR |
| **Clavijo et al., 2003** [56] | Spain | Culture e/ ou SAT | 1) Rose Bengal; 2) ELISA; 3) Rapid test | 133 | Retrospective | < 3 (n = 87) / > 3 (n = 46) | 42 (16–75) | 88/22 | NR |
| **Aranís et al., 2008** [57] | Chile | Culture and/or SAT | 1) Coombs; 2) ELISA | 10/18 | Prospective | 0.6 (0.2–1) | M:53; F:46 | 6/4 | NR |
| **Gómez e tal., 2008** [58] | NR | Culture and/or SAT | 1) Rose Bengal; 3) ELISA; 4) Coombs | 25/90 | Retrospective | NR | 41 (12–80) | 22:3 | NR |
| **Hasibi et al., 2008** [59] | Iran | Culture and/or SAT | 1) ELISA; 2) PCR | 37/78 | Prospective | NR | 44.8±14.7 | Case: 57/43; Control: 91/9 | NR |
| **Fadeel et al., 2011** [60] | Egypt e USA | Culture and/or SAT | ELISA | 186/183 | Retrospective | NR | NR | NR | NR |
| **Marei et al., 2011** [61] | Egypt | Culture and/or SAT | 1) Rose Bengal; 2) PCR; 3) Rapid test | 20/30 | Prospective | 0.23 a 1.4 | 32.9 (21–63) | 13/7 | NR |
| **Hasibi et al., 2013** [62] | Iran | Culture and/or SAT | ELISA | 56/126 | Retrospective | NR | NR | NR | *B. melitensis* (19) |

SAT–standard tube agglutination test; ELISA–immunoenzymatic assay; PCR–polymerase chain reaction; SD–standard deviation; NR–not reported; IQ–interquartile range.

(Fig 6) [53,55,59,61]. In most studies, B4 and B5 primers were used to detect the BCSP31 gene in whole blood samples. It is worth noting that one study targeted the 16S RNA gene [53], and in another study, the clinical specimen was serum [61]. In one study [59] a high detection limit was reported (1000fg), leading to a low sensitivity (45.5%). The exclusion of this study [59] from analyses promoted a reduction in the observed heterogeneity, with a sensitivity of 93.3% [95%CI: 81.4–97.8; $i^2$ = 24.9] and specificity of 86.1% [95%CI: 80.0–90.5; $i^2$ = 0.00]. No study addressing real-time PCR with culture and/or SAT as a reference test was found.

## Risk of bias assessment

In general, a high risk of bias was observed for patient selection, reference standard, and flow and timing (Fig 7). For patient selection, the high risk of bias was mainly associated with the use of historical sample panels without clear inclusion and exclusion criteria. For the reference standard, all studies were considered to have an imperfect reference standard. In the flow and

**Table 2. Pooled performance measure of diagnostic tests evaluated for human brucellosis, using both reference tests.**

| | Culture | | | | Culture and/or SAT | | | |
|---|---|---|---|---|---|---|---|---|
| | Sensitivity (%) [95%CI] | I² | Specificity (%) [95%CI] | I² | Sensitivity (%) [95%CI] | I² | Specificity (%) [95%CI] | I² |
| **Serological test** | | | | | | | | |
| Rose Bengal | 89.7 [82.0–94.4] | 70.8 | 94.1 [83.1–98.1] | 90.3 | 96.6 [92.6–98.5] | 0.00 | 97.9 [93.1–99.4] | 0.00 |
| SAT | 89.2 [81.4–94.0] | 85.2 | 95.6 [89.7–98.1] | 89.2 | - | - | - | - |
| IgG ELISA | 82.9 [59.5–94.1] | 92.7 | 96.2 [80.6–99.4] | 86.1 | 85.8 [75.5–92.2] | 0.00 | 99.0 [95.3–99.8] | 0.00 |
| IgM ELISA | 84.5 [68.0–93.3] | 86.5 | 95.3 [87.5–98.4] | 56.3 | 55.3 [47.8–62.7] | 0.00 | 96.8 [59.5–99.8] | 73.5 |
| IgA ELISA | 94.4 [68.1–99.2] | 77.6 | 98.5 [90.3–99.8] | 0.00 | - | - | - | - |
| IgG/IgM ELISA | 94.3 [87.5–97.5] | 68.6 | 93.5 [75.8–98.5] | 94.3 | 96.8 [60.8–99.8] | 88.9 | 98.6 [96.1–99.5] | 0.00 |
| IgG ICT test | 78.7 [70.0–85.4] | 0.00 | - | - | - | - | - | - |
| IgM ICT test | 74.3 [35.2–93.9] | 91.9 | - | - | 70.6 [62.9–77.3] | 0.00 | - | - |
| IgG/IgM ICT test | 96.2 [70.4–99.6] | 63.9 | - | - | - | - | - | - |
| Coombs | 93.1 [60.2–99.2] | 71.5 | 98.4 [91.0–99.7] | 18.9 | 89.4 [30.0–99.4] | 73.6 | 98.8 [92.0–99.8] | 0.00 |
| **Molecular test** | | | | | | | | |
| Qualitativel PCR | 96.4 [69.6–. 99.7] | 82.2 | 98.1 [93.6–99.5] | 30.7 | 79.6 [47.6–94.4] | 90.4 | 96.0 [85.8 – 99.0] | 47.8 |
| Real time PCR | 81.9 [66.9–91.0] | 74.1 | 91.5 [71.4–97.9] | 89.3 | - | - | - | - |

ICT- immunochromatographic test; SAT–standard tube agglutination test; ELISA–immunoenzymatic assay; PCR–polymerase chain reaction.

timing domain, the high risk of bias arises from the use of different reference standards for confirming or ruling out diagnoses, with healthy patients often included for specificity analysis. Concerning the index test, several studies do not clearly state whether the index tests were interpreted without knowledge of the reference standard result, and in some studies, the cut-off values were not pre-specified.

Regarding the applicability of the articles to the review question, a high level of concern was noted, primarily related to patient selection. This was particularly relevant for studies that used healthy patients as controls or those without a non-case group.

### Certainty of evidence

The certainty of the evidence was assessed using the GRADE system for the following tests: Rose Bengal, IgG/IgM ELISA, and qualitative PCR, considering culture and/or SAT as the comparator. In all comparisons, the certainty of the evidence was classified as very low, mainly due to penalties in the areas of risk of bias, indirect evidence, imprecision, and publication bias (S4 File).

### Discussion

The diagnosis of brucellosis has been a focus of recent narrative reviews and research [10,11,63]. Although, to our knowledge, this is the first systematic review providing a

**Sensitivity:** Rose Bengal x Culture and/or SAT

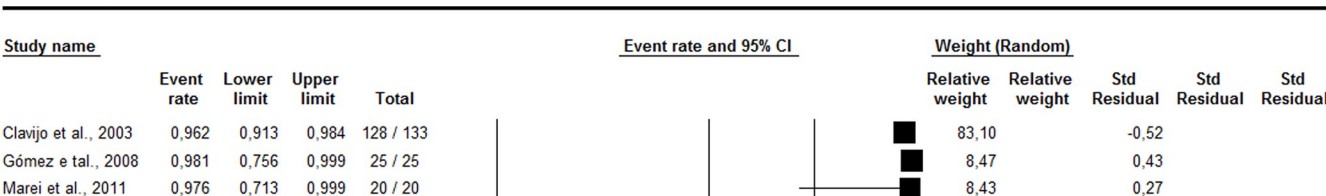

Heterogeneity: I-square=00.00, tau-squared=0.00, p=0.87

**Specificity:** Rose Bengal x Culture and/SAT

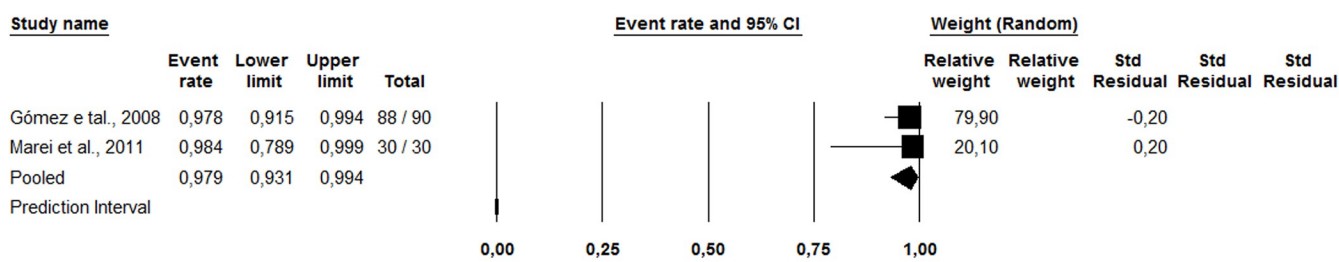

Heterogeneity: I-square=00.00, tau-squared=0.00, p=0.84

**Fig 2. Rose Bengal's sensitivity and specificity for human brucellosis diagnosis considering culture and/or SAT as the reference test.**

comprehensive overview of diagnostic methods for brucellosis. Here, some important aspects can be highlighted: (i) the available evidence is limited, and significant variability exists among studies; (ii) cultures combined with SAT seem to provide more appropriate reference standards than culture alone; (iii) Rose Bengal, IgG/IgM ELISA, and PCR exhibited equally high

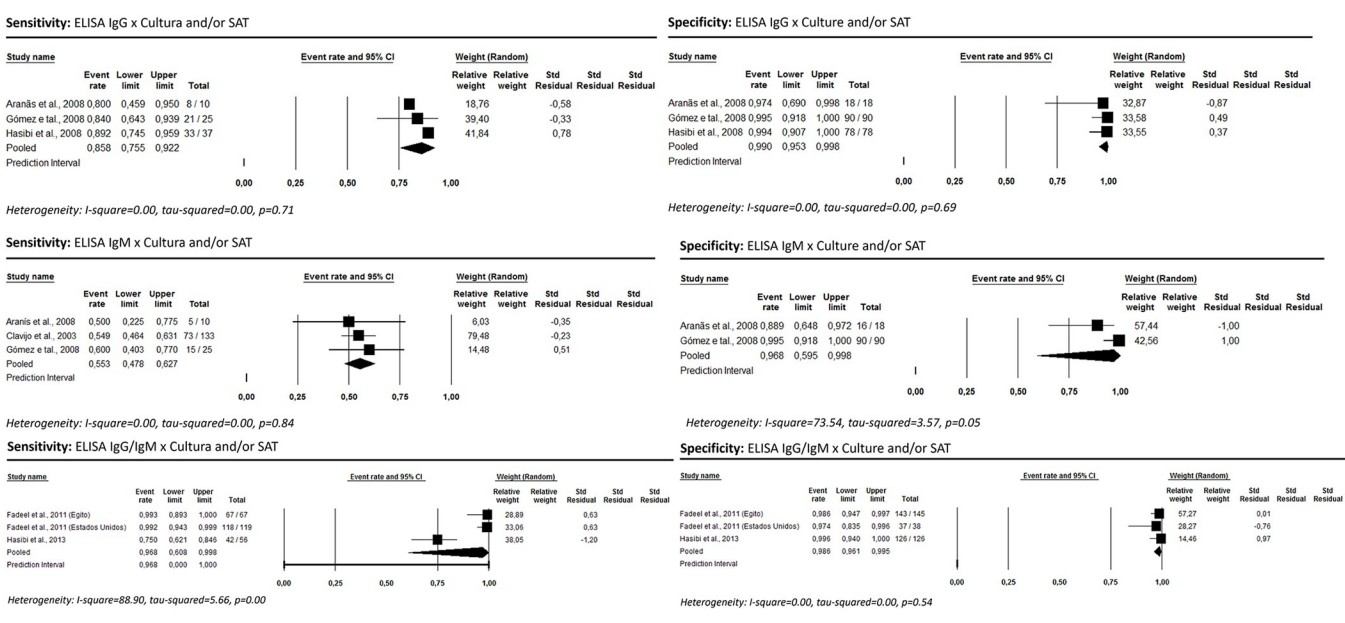

**Fig 3. ELISA's sensitivity and specificity for human brucellosis diagnosis considering culture and/or SAT as reference test.**

**Sensitivity :** IgM Rapid diagnostic test x Culture and/or SAT

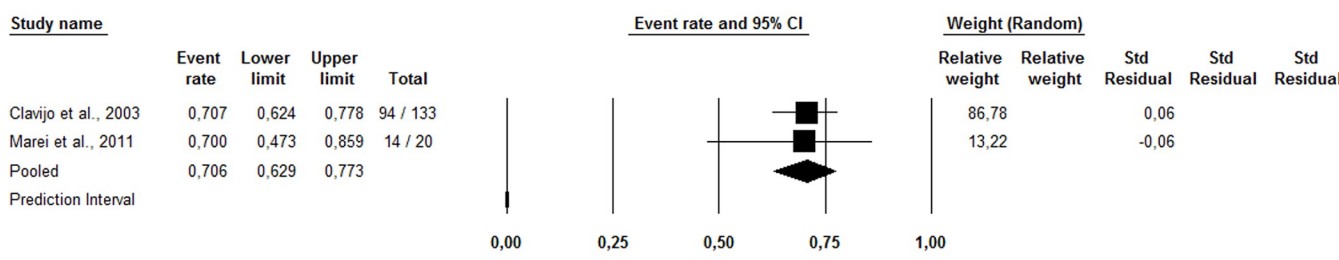

Fig 4. **Immunochromatography's sensitivity for human brucellosis diagnosis using culture combined with the SAT as a reference test.**

performances. However, caution is required when interpreting the results due to the heterogeneity in the design of the studies, the patient profiles, and the methodological weaknesses of the assessed studies.

Three diagnostic tests stood out in this review, either due to the frequency with which they were reported or their high performance. These were the Rose Bengal, IgG/IgM ELISA, and qualitative PCR tests. It was not possible to differentiate between these tests by analyzing the pooled sensitivities and specificities due to the overlapping confidence intervals. The main concern regarding these high performances was the evident case selection bias, as most studies were retrospective series [26–31,33,34,36–40,42–45,48,51,54,56,58,60,62]. In addition to intrinsic sources of bias, such as the overestimation of the performance of the tests, other aspects related to operability, cost, and test requirements must be considered when conducting a comparison [3]. This is especially important, considering brucellosis mainly affects people in

**Sensibilidade:** Coombs X Cultura e/ou SAT

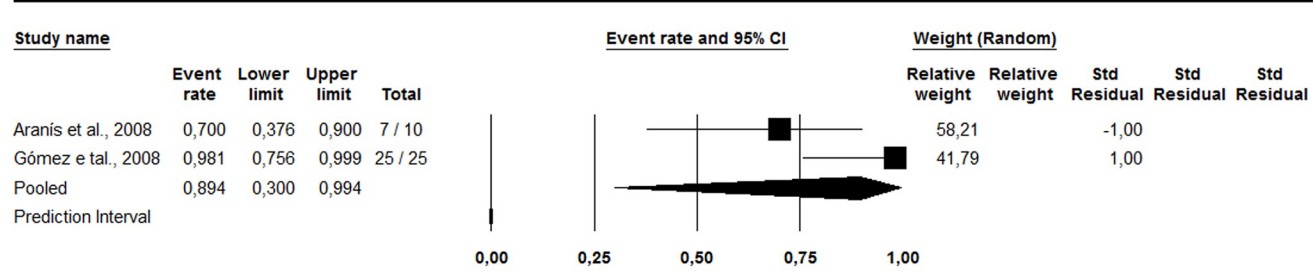

**Specificity :** Coombs X Culture and/or SAT

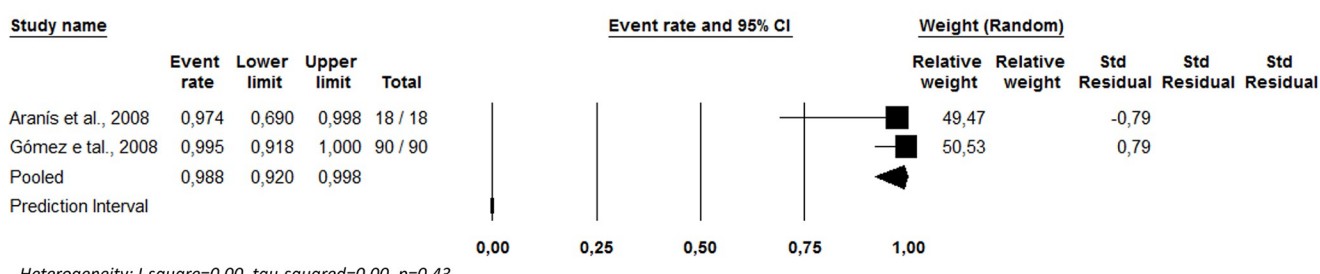

Fig 5. **Coombs tests' sensitivity and specificity for human brucellosis diagnosis using culture combined with the SAT as a reference test.**

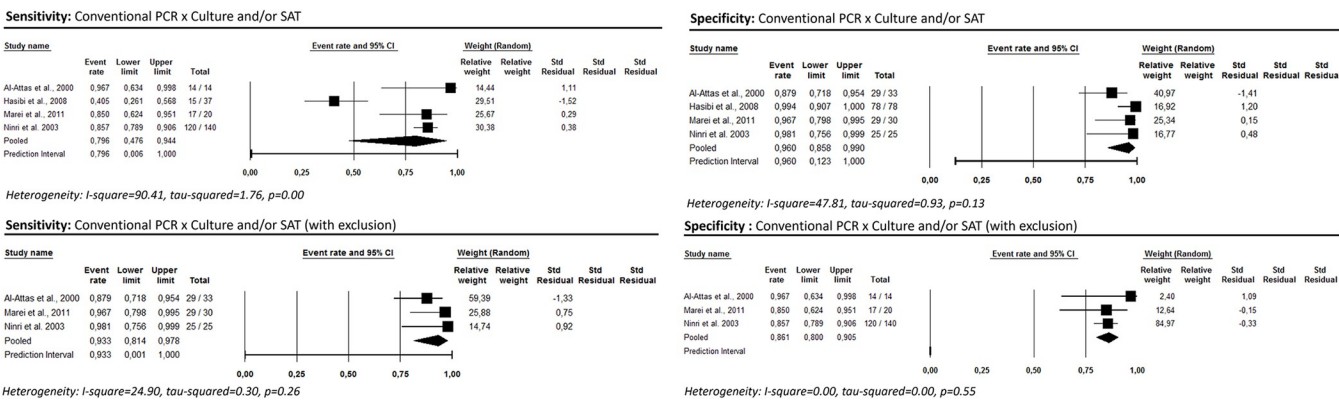

**Fig 6. Qualitative PCR's sensitivity and specificity for human brucellosis diagnosis considering culture and/or SAT as reference test.**

resource-limited settings [64]. It is therefore worth considering that the Rose Bengal test is relatively straightforward and cost-effective, making it particularly valuable in regions where human brucellosis is endemic and where laboratory facilities and resources are limited. ELISA tests have the advantage that they are widely commercially available and come with a high degree of automation, making them suitable for evaluating a larger number of patients. On the other hand, PCR is a method that still lacks standardization—different targets and protocols were used in the studies, and prior validation of the procedure is essential to ensure an acceptable detection limit [30]. Additionally, PCR tests require specialized equipment and knowledge, which is particularly difficult to obtain in the regions where brucellosis is prevalent [15,65].

Considering the significant differences between the agglutination, immunoassay, and PCR methods, the similar high performances they achieved merit a detailed analysis to investigate any potential biases that could have influenced them. The high performance observed for the Rose Bengal test could be due to a sample selection, with patients recruited during the early stages of the disease, typically within three months of symptom onset [56,58,61]. However, a single study assessing the test's performance as a function of the disease duration did not find

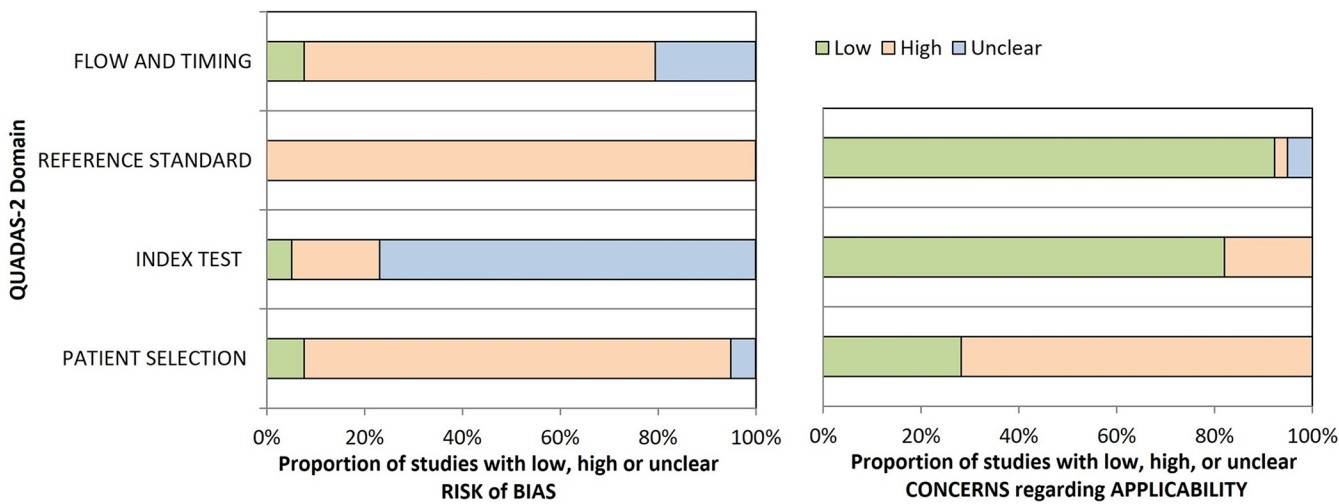

**Fig 7. Risk of bias assessment by the Quadas-2 tool for diagnostic accuracy studies.**

any significant differences between the groups [56]. For the ELISA tests, it is important to note that a poor description was given of antigens used, both for *B. melitensis* [32] and *B. abortus* [31,34,41,56,59] in many studies. Most of the discussed ELISA tests are commercially available (88%), but in-house ELISA tests were also used [26,32]. In many studies, healthy controls were used [33,54,56], and in others, patients were evaluated at different stages of the disease. An example of this is the study performed by Araj et al. (1988) [32], in which patients with both the acute and chronic phases were discussed. All these factors could potentially influence the results. It is somewhat surprising that PCR does not show superiority over serological tests, contrary to the diagnostic algorithms proposed by certain health authorities [64,66]. The main suspected reason for the relatively low PCR sensitivity is the phase of the disease when the test was performed, as there may have been a low or intermediate bacteremia load at different stages of the disease. The high sensitivity of PCR tests was observed in studies that assessed patients during the first weeks of symptoms [35,46,56]. However, the duration of symptoms is lacking in many studies [35,36,42,51,59]. Consistent with this observation, several studies imply that the reduced sensitivity in PCR compared to serological tests could be associated with cases where patients received inadequate prior antibiotic treatment [30,67,68]. Marei et al. 2011 demonstrated a decline in PCR sensitivity from 85% to 31% in pre- and post-treatment blood samples [61]. This indirect evidence suggests that molecular tests might be more suitable for investigating patients within the first weeks of symptom onset, presumably during the bacteremia phase. Even if the high risk of bias affected the tests' performance, using the same comparator was used to attempt to align the studies. The present study confirms the need for more investment in supporting both the development of new diagnostic tests and well-designed studies for brucellosis [63].

In addition, another factor to be considered when evaluating the accuracy of the tests is the 'threshold effect,' which derives from the interrelationship between sensitivity and specificity; in other words, as sensitivity increases, specificity typically decreases. This effect may result from explicit variations in positive cut-off definitions or implicit differences in study populations and methodologies [69]. Such a phenomenon was evident for two of the evaluated Rose Bengal [25,52] tests and one IgG/IgM ELISA test [24] showed notably low specificity alongside high sensitivities.

In terms of specificity, various factors can affect its value. Culture, in comparison to other tests, demonstrates low sensitivity, leading to potentially lower specificity rates when used as a reference standard. Additionally, including healthy individuals in the control group poses a significant risk of data overestimation, as this group is not suitable for the index test. It is crucial to incorporate patients affected by other infectious diseases that clinically resemble brucellosis, such as typhoid fever, bacterial endocarditis, tuberculosis, and malaria, to assess cross-reactivity with *Brucella*. Moreover, the characteristics of the antigen, such as purity and type (purified, recombinant, or synthetic peptides), constitute another important factor influencing specificity assessment in the control group.

The definition of reference standards plays a crucial role in diagnostic accuracy studies, directly impacting test performance. The absence of a universally acknowledged gold standard test often presents a challenge, potentially leading to overestimation or underestimation of sensitivity and specificity rates. This variability depends on the frequency of classification errors made by the reference standard and the degree of correlation of errors between the index test and the reference standard [70]. In the context of brucellosis, culture is frequently used as a reference standard due to its high specificity. However, the variable and generally lower sensitivity of culture significantly impacts the accuracy of index tests. Our observation, when using culture as a reference test, revealed considerable heterogeneity in summary measurements and lower specificity of the index tests, likely stemming from the inherent low sensitivity of culture.

To address this limitation, we adopted a reference standard of 'culture and/or SAT,' resulting in reduced heterogeneity and seemingly fewer flaws in the obtained results.

Progress in diagnosing brucellosis has been limited, reflecting the underinvestment in this disease. A validated diagnostic strategy or a widely used point-of-care test is still lacking. Although some validation studies addressing rapid tests have been identified in the literature, a notable risk of bias was found in them, mainly due to the lack of specificity data [25,36,51], and the accuracy was, in general, low. Furthermore, the limited availability of commercial rapid tests prevents both its proper validation and its recommendation for large-scale use currently. In short, the available data are insufficient to explain the poor performance of rapid tests, which could be related to the antigen used or to the lower detection threshold used in immunochromatography. Another approach for improving the accuracy of the rapid tests would be developing a test based on the simultaneous detection of IgG and IgM antibodies, an apparently successful strategy for the ELISA platform. Further research and investment are required to develop more robust and accessible diagnostic tools for brucellosis. In the future, it is possible that the exploration of immune responses could play a pivotal role in advancing the development of innovative diagnostic tools for brucellosis. The identification of key elements in these responses, such as cytokines, antibodies, and cellular reactions, in this thesis, could contribute to the development of diagnostic assays incorporating multiple markers, enabling a more comprehensive and reliable diagnosis of brucellosis [71,72].

Our analyses confirm significant heterogeneity in sensitivity and specificity estimates, especially when culture is used as the reference standard. This raises important questions about the reliability and consistency of these tests in various clinical and epidemiological settings. While the reasons for this heterogeneity may vary, factors such as the variable and generally low sensitivity of culture, variations in patient populations, diversity of *Brucella* species, and test protocols should be carefully considered [7,8]. Clinicians should be aware of the potential variation in test performance and consider it when interpreting results, especially in regions with diverse epidemiological profiles.

The included studies exhibit a high risk of bias in domains related to patient selection, reference standards, and flow and timing, which can significantly affect the validity of reported diagnostic accuracy estimates [23]. Using an imperfect reference standard and including healthy individuals as controls can introduce significant challenges when interpreting test performance. Essential information is often lacking in study reports, such as inclusion and exclusion criteria adopted in the primary study, time of symptom onset, disease severity, and clinical classification. In bacterial diseases like brucellosis, the concentration of antibodies and the presence of agent DNA in patient samples could vary throughout the disease [10]. Thus, the lack of complete characterization and criteria for selecting cases and classifying disease status are the main factors hindering the critical use of the information provided by studies. Tools like reporting of diagnostic accuracy studies (STARD) must be employed to enhance the quality of reporting in diagnostic accuracy studies, providing the essential aspects that must be presented [73].

The high number of studies on Brucellosis in the Asian region is noteworthy, primarily attributed to the heightened prevalence of the disease in countries such as Turkey, Saudi Arabia, and Kuwait. Indeed, caution is necessary to extrapolating findings to other geographical areas. Variability in the performance of diagnostic tests for infectious diseases across regions requires careful consideration. Factors such as disease prevalence, the existence of diverse strains, and variations in the expertise of healthcare professionals may interfere with the interpretation of the results. Therefore, the use of results from other global contexts demands a thoughtful.

Due to insufficient information in the original studies, we could not stratify diagnostic test performance based on the disease's stages. The timing of diagnostic testing holds significance

not only for individual patient management but also for public health initiatives, including contact tracing and outbreak control. Ensuring accurate and timely diagnosis is paramount in preventing the spread of brucellosis, whether in localized cases or larger-scale outbreaks. Consequently, future research endeavors should consider conducting longitudinal studies that capture the disease's dynamics, encompassing variations in the immune response and bacterial load over time.

The primary limitations of the systematic review stem from the scarcity and quality of the studies included. Due to a limited number of available studies and insufficient data from a real control group (individuals displaying symptoms resembling brucellosis but diagnosed with a different disease), information from all identified studies was incorporated, even if sensitivity and specificity were not jointly presented. Specificity data often originates from patients unsuitable for practical diagnosis, potentially leading to overestimation. The limitations of available data also impeded the stratification of critical factors that could influence the test accuracy, such as variations in *Brucella* species and infection stage. The review depended on limited evidence, with no substantial, well-designed prospective studies evaluating serological or molecular tests documented thus far. Furthermore, the existing data inadequately represents all endemic regions. While awaiting more comprehensive investigations for a deeper understanding, healthcare professionals and endemic countries should consider all available information and factors beyond accuracy in decision-making.

In conclusion, our systematic review sheds light on the current state of diagnostic methods for brucellosis. Despite the substantial efforts, several challenges and uncertainties remain in the field of brucellosis diagnosis. The limitations of available evidence, coupled with significant variability between studies, underline the need for further research and standardization of diagnostic protocols. Our results confirm the usefulness of Rose Bengal, IgG/IgM ELISA, and PCR. However, although these tests seem similar in accuracy, their applicability may vary depending on the local context and available resources, in addition to the stages of the disease, which still require better-designed studies. The accessibility, affordability, and scalability of these diagnostic methods must be considered to ensure equitable healthcare. As we strive for more accurate, accessible, and context-specific diagnostic methods, collaboration among researchers, healthcare providers, and managers remains crucial.

## Supporting information

**S1 File. Search strategy used in each database.**
(DOCX)

**S2 File. Detail of each index test according to the included studies.**
(DOCX)

**S3 File. Sensitivity and specificity of all evaluated index tests for diagnosing human brucellosis, using culture as reference standards.**
(DOCX)

**S4 File. Evaluation of the certainty of evidence using the GRADE system.**
(DOCX)

## Acknowledgments

We are grateful for all the support and endorsement of the Brazilian Ministry of Health, specifically the General Coordination of Management of Clinical Protocols and Therapeutic Guidelines (CGPCDT) and Instituto René Rachou (Fundação Oswaldo Cruz).

## Author Contributions

**Conceptualization:** Mariana Lourenço Freire, Sarah Nascimento Silva, Gláucia Cota.

**Data curation:** Mariana Lourenço Freire, Tália Santana Machado de Assis, Gláucia Cota.

**Formal analysis:** Mariana Lourenço Freire, Gláucia Cota.

**Investigation:** Mariana Lourenço Freire, Tália Santana Machado de Assis.

**Methodology:** Mariana Lourenço Freire, Tália Santana Machado de Assis, Gláucia Cota.

**Resources:** Gláucia Cota.

**Validation:** Mariana Lourenço Freire, Tália Santana Machado de Assis, Gláucia Cota.

**Writing – original draft:** Mariana Lourenço Freire, Tália Santana Machado de Assis.

**Writing – review & editing:** Mariana Lourenço Freire, Tália Santana Machado de Assis, Sarah Nascimento Silva, Gláucia Cota.

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
