## [Decision Letter · Decision Letter 0]

14 Jan 2024

Dear Dr. Freire,

Thank you very much for submitting your manuscript "DIAGNOSIS OF HUMAN BRUCELLOSIS: SYSTEMATIC REVIEW AND META-ANALYSIS" for consideration at PLOS Neglected Tropical Diseases. As with all papers reviewed by the journal, your manuscript was reviewed by members of the editorial board and by several independent reviewers. In light of the reviews (below this email), we would like to invite the resubmission of a significantly-revised version that takes into account the reviewers' comments.

Please respond to the reviewer comments thoroughly in a point-by-point response.

We cannot make any decision about publication until we have seen the revised manuscript and your response to the reviewers' comments. Your revised manuscript is also likely to be sent to reviewers for further evaluation.

Sincerely,

Georgios Pappas

Academic Editor

Justin Remais

Section Editor

Please respond to the reviewer comments as thoroughly as needed.

Reviewer's Responses to Questions

**Key Review Criteria Required for Acceptance?**

**Methods**

-Are the objectives of the study clearly articulated with a clear testable hypothesis stated?

-Is the study design appropriate to address the stated objectives?

-Is the population clearly described and appropriate for the hypothesis being tested?

-Is the sample size sufficient to ensure adequate power to address the hypothesis being tested?

-Were correct statistical analysis used to support conclusions?

-Are there concerns about ethical or regulatory requirements being met?

Reviewer #1: (No Response)

Reviewer #2: (No Response)

**Results**

-Does the analysis presented match the analysis plan?

-Are the results clearly and completely presented?

-Are the figures (Tables, Images) of sufficient quality for clarity?

Reviewer #1: (No Response)

Reviewer #2: (No Response)

**Conclusions**

-Are the conclusions supported by the data presented?

-Are the limitations of analysis clearly described?

-Do the authors discuss how these data can be helpful to advance our understanding of the topic under study?

-Is public health relevance addressed?

Reviewer #1: (No Response)

Reviewer #2: (No Response)

**Editorial and Data Presentation Modifications?**

Reviewer #1: (No Response)

Reviewer #2: (No Response)

**Summary and General Comments**

Reviewer #1: The article by Mariana Lourenço Freire et al., about diagnosis is interesting and informative. However, a few comments have to be addressed before publication.

• Despite the review on the diagnosis of brucellosis in humans, however, a broad introduction to the challenges of diagnosis in general and the most recently used technology in diagnosis have to be included; for example, check and cite the following articles: Doi: https://doi.org/10.51585/gjvr.2023.3.0056, doi: 10.3389/fmicb.2023.1259479. eCollection 2023. 

• Change gram-negative into Gram, line 64

• Information in table 1 is arranged based on the authors' names alphabetically; I suggest arranging based on the date of study from recent to the oldest, or based on countries or even the test used to be more practical and informed.

• I suggest dividing the results into serology and molecular or culture-based.

• In the discussion, the authors mentioned that “To our knowledge, this is the first systematic literature review of diagnostic methods for human brucellosis,” which is an overestimation as there are some studies that discussed the same issue before. For example, this chapter, which is available at https://doi.org/10.1007/978-981-13-8844-6_16

• Please refer in discussion to the several immune responses and recent advances in the diagnosis and control of brucellosis, which have been published at https://doi.org/10.51585/gjvr.2022.1.0033

Reviewer #2: Q1. Overall, your introductory section provides a detailed description of the background information on brucellosis and the direction and purpose of the study. Although various existing diagnostic methods were described, a clear description of why this study was conducted was lacking. Is it that existing diagnostic methods are not sufficient to meet the actual needs, or are there potentially more efficient diagnostic tools that have not yet been widely adopted? The last sentence clearly states the purpose of the study, but it could be further clarified as to how the expected results will solve existing problems or improve the current diagnostic process.

Q2. Although it is mentioned that keywords were used for the search, no specific list of keywords or search equations are provided in this section. Providing a complete search strategy would help others replicate the findings and enhance the transparency of the study.

Q3. Have you conducted sensitivity analyses after excluding each sample separately, and what was the effect of each exclusion on the overall outcome indicator, please provide an overview in the text.

Q4. You need to have a discussion about the inclusion of healthy individuals as a control group, explaining how this practice affects the interpretation of diagnostic test performance.

Q5. In the discussion, please further elaborate on the specific problem of incomplete reference standards and their potential impact on the study's conclusions.

Q6. If possible, explore the limitations that geographical constraints may impose on the study.

PLOS authors have the option to publish the peer review history of their article (what does this mean?). If published, this will include your full peer review and any attached files.

Reviewer #1: No

Reviewer #2: No
---

## [Decision Letter · Decision Letter 1]

16 Feb 2024

Dear Dr. Freire,

Thank you very much for submitting your manuscript "DIAGNOSIS OF HUMAN BRUCELLOSIS: SYSTEMATIC REVIEW AND META-ANALYSIS" for consideration at PLOS Neglected Tropical Diseases. As with all papers reviewed by the journal, your manuscript was reviewed by members of the editorial board and by several independent reviewers. The reviewers appreciated the attention to an important topic. Based on the reviews, we are likely to accept this manuscript for publication, providing that you modify the manuscript according to the review recommendations. 

Please respond to reviewer comments in a point-by-point response.

Sincerely,

Georgios Pappas

Academic Editor

Justin Remais

Section Editor

please respond to these comments

Reviewer's Responses to Questions

**Key Review Criteria Required for Acceptance?**

**Methods**

-Are the objectives of the study clearly articulated with a clear testable hypothesis stated?

-Is the study design appropriate to address the stated objectives?

-Is the population clearly described and appropriate for the hypothesis being tested?

-Is the sample size sufficient to ensure adequate power to address the hypothesis being tested?

-Were correct statistical analysis used to support conclusions?

-Are there concerns about ethical or regulatory requirements being met?

Reviewer #1: objectives of the study clearly provided and the authors followed PRISMA guidlines

**Results**

-Does the analysis presented match the analysis plan?

-Are the results clearly and completely presented?

-Are the figures (Tables, Images) of sufficient quality for clarity?

Reviewer #1: The results have been provided clearly

**Conclusions**

-Are the conclusions supported by the data presented?

-Are the limitations of analysis clearly described?

-Do the authors discuss how these data can be helpful to advance our understanding of the topic under study?

-Is public health relevance addressed?

Reviewer #1: the authors provided clear conclusion supporting the data anlysed

**Editorial and Data Presentation Modifications?**

Reviewer #1: (No Response)

**Summary and General Comments**

Reviewer #1: This systematic review and meta-analysis on the diagnosis of human brucellosis is very interesting, well-written, informative, and suitable for publication in this journal. However, the methodology needs to be included in the abstract, despite being well-written in the method section. Please provide a brief explanation of how and when the data was collected, including the criteria for collecting data, the time period, and any exclusion criteria used. The rest of the manuscript is acceptable and is well-supported by enough figures and tables."

PLOS authors have the option to publish the peer review history of their article (what does this mean?). If published, this will include your full peer review and any attached files.

Reviewer #1: No

Figure Files:

Data Requirements:

Reproducibility:

References

---

## [Editor Report · Decision Letter 2]

27 Feb 2024

Dear Dr. Freire,

We are pleased to inform you that your manuscript 'DIAGNOSIS OF HUMAN BRUCELLOSIS: SYSTEMATIC REVIEW AND META-ANALYSIS' has been provisionally accepted for publication in PLOS Neglected Tropical Diseases.

Best regards,

Georgios Pappas

Academic Editor

Justin Remais

Section Editor

-

---

## [Editor Report · Acceptance letter]

4 Mar 2024

Dear Dr. Freire,

We are delighted to inform you that your manuscript, "DIAGNOSIS OF HUMAN BRUCELLOSIS: SYSTEMATIC REVIEW AND META-ANALYSIS," has been formally accepted for publication in PLOS Neglected Tropical Diseases.

Best regards,

Shaden Kamhawi

co-Editor-in-Chief

Paul Brindley

co-Editor-in-Chief
